# Morphology of the light-organ system and bioluminescent blinking in the ponyfish tribe Equulitini (Teleostei: Leiognathidae)

**Michael J. Ghedotti**[1,2]*, **Jordon J. Valdez**[1], **Rene P. Martin**[3], **Emily M. Carr**[4], **John S. Sparks**[4]

**1** Biology, Regis University, Denver, Colorado, United States of America, **2** Bell Museum of Natural History, University of Minnesota, St. Paul, Minnesota, United States of America, **3** School of Natural Resources, University of Nebraska, Lincoln, Nebraska, United States of America, **4** Department of Ichthyology and Richard Gilder Graduate School, American Museum of Natural History, New York, New York, United States of America

* mghedott@regis.edu

## Abstract

Ponyfishes in the Family Leiognathidae uniquely possess a complex bioluminescent system with an esophageal light-producing organ containing bioluminescent bacterial symbionts, a reflective gas bladder, and transparent windows through adjacent tissues that allow light emission distant from the esophageal source to produce camouflaging counterillumination and, in some species, intraspecific communication and possibly illumination of prey. Despite having bacterial light production, a type of light production that in fishes is not known to be able to be rapidly initiated and terminated, researchers have observed many species of ponyfishes rapidly blinking. The tribe Equulitini is a well diagnosed clade of sexually dimorphic ponyfishes in which males have lateral transparent windows on the body, and males of one species, *Equulites elongatus,* were observed in the wild to blink rapidly through their lateral windows. This group allowed us to explore bioluminescent organ anatomy in a sexually dimorphic group and describe the morphology associated with blinking. In this study we used gross anatomical, histological, and radiological methods to investigate and describe the light organ-system of ponyfishes in the Tribe Equulitini. Although previously described generally, we provide the first detailed description of the internal structure of the bioluminescent light organ in Leiognathidae, identify the esophageal homologs of the light organ's tissues, and describe the novel mechanism for blinking. Blinking occurs when transparent longitudinal muscle that extends over the dorsal and ventral surfaces of the light organ contract or relax extending or retracting mucosal connective tissue shutters with respect to the light emitting transparent windows on the light organ. Although males have larger light organs and transparent lateral windows on their flanks, both males and females have a similar morphology and have structures consistent with the ability to blink. We also discuss variation in bioluminescent anatomy within Tribe Equulitini.

**Data availability statement:** All specimens used in this study are available from their indicated museums, and histological slides are deposited in the Ichthyology collection at the American Museum of Natural History (https://www.amnh.org/research/vertebrate-zoology/ichthyology/collection-information), and µCT scan data are available at Zenodo (DOI: 10.5281/zenodo.19295041).

**Funding:** The funders Regis University and the American Museum of Natural History provided support in the form of salaries for authors M.J.G., E.M.C., J.S.S., and a Regis University URSC Faculty Grant (URSCMJG2025) to M.J.G. https://one.regis.edu/academics/research-grants/ursc provided funds for supplies. The funders played no role in the study design, data collection and analysis, decision to publish, or preparation of the manuscript.

**Competing interests:** The authors have declared that no competing interests exist.

## Introduction

Fishes in the Family Leiognathidae, commonly known as ponyfishes or slipmouths, are unique for their possession of a complex bioluminescent system that includes an esophageal light-producing organ, reflective tissues in the gas bladder, and transparent windows through adjacent tissues that result in light emission at surface locations distant from the esophagus [1–4]. Ponyfish light organs house symbiotic, bioluminescent bacteria of the genera *Photobacterium* and *Vibrio* in chambers connected to the central cavity of the esophagus [5–8]. Ponyfishes are schooling fishes that commonly inhabit turbid nearshore marine and estuarine habitats with low visibility where it is thought that their light emission facilitates both camouflage and communication and may also allow illumination of prey [9–11].

All ponyfishes regardless of sex are thought to be able to emit light from their ventral surfaces as a means of counterillumination in turbid waters [9,11,12]. Light from the esophageal light organ is transmitted directly through transparent pectoral tissues and posteriorly to the reflective guanine-lined gas bladder. The reflected light in the gas bladder exits the body through posteroventral transparent tissues to generate posteroventral counterillumination [1,2]. Although the ventral surface appears reflectively opaque, gaps between the dorsoventrally positioned guanine crystals in the dermis may reflect lateral light while allowing light to exit ventrally as have been demonstrated in ponyfishes and other groups [2,13]. Higher ventral light emission intensity occurs in species found at greater depths and in more turbid habitats where a silhouette would be more visible to predators, consistent with use of ventral counterillumination as silhouette camouflage [14].

Many species of ponyfishes have sexually dimorphic bioluminescent systems that suggest a role for light production in sexual display [5,12,15,16,17]. In many species males have larger esophageal light organs than females and/or transparent light windows that are either lacking in conspecific females, or much less well developed [10], which suggests sex-specific communication. There is a close association between development of dimorphism and reproductive development in the offshore ponyfish *Equulites rivulatus*, where greater male light organ growth is closely coupled with gonadal maturation [18]. Chakrabarty et al. [19] found no evidence for increased diversifying sexual selection given that species diversity and body-shape disparity is similar when compared between dimorphic and non-dimorphic ponyfishes. Davis et al. [20] report a speciation rate for Leiognathidae consistent with a higher rate of diversification that could be driven by sexual selection. However, sexual signaling may but does not necessarily result in diversifying sexual selection, and ponyfish light-organ dimorphism strongly suggests sexual communication in dimorphic ponyfishes.

Rapid blinking has been observed in species of ponyfishes within the genera *Deveximentum*, *Equulites*, *Eubleekeria*, *Gazza*, *Nuchequula*, and *Photopectoralis* wherein emitted bioluminescence rapidly appears and/or rapidly dims within a one second period or less [2,5,21,22,23,24]. In some ponyfishes this blinking behavior has been observed to occur in both males and females and may include ventral, lateral, and cephalic light emission [2]. Because the luminescent blinking observed

in ponyfishes occurs rapidly it is unlikely to be caused by expansion and contraction of dark chromatophores and iridophores, and instead likely occurs via the action of skeletal muscle on light-screening tissues [2,5,25,22,26].

The esophageal light organ of ponyfishes develops as a dorsal diverticulum of the esophagus immediately anterior to the diverticulum of the developing gas bladder and is colonized by bioluminescent bacteria shortly after development begins [7]. The light organ grows from the initial dorsal diverticulum to cover the dorsal and lateral surfaces of the esophagus and in most species rings the esophagus, growing to fully surround the esophagus ventrally [5,7,27,25]. The light organ is screened by dark chromatophores (likely melanophores) and iridophores containing guanine, as well as by cells or cell clusters containing both guanine and dark pigment from early in development, limiting the emission of light into surrounding tissues [7,25,27,28]. The fully developed light organ has anterior and posterodorsal windows in the tissues otherwise screened by continuous guanine and/or dark chromatophores and iridophores, which allow light to enter the transparent pectoral tissues and the reflective posterodorsal gas bladder that is lined with silvery reflective guanine, except in specific regions (e.g., posteroventrally and laterally) where light is emitted [5,21,22].

Leiognathidae includes 53 species in 10 genera and is distributed throughout the tropical and subtropical Indo-West Pacific, including estuarine habitats [29–31]. Leiognathidae is the sister taxon to the worldwide tropical and subtropical marine butterflyfishes, family Chaetodontidae [14,32]. Butterflyfishes are commonly associated with reef habitats and clear water, whereas ponyfishes are more commonly associated with estuarine habitats and higher turbidity [14,31]. The subfamily Leiognathinae contains the non-sexually dimorphic genera *Leiognathus* and *Aurigequula* and is variously recognized as either the monophyletic sister taxon to or a paraphyletic grade with respect to the remaining sexually dimorphic ponyfishes in the monophyletic subfamily Gazzinae [15,19,30,33]. Within Gazzinae, the tribe Equulitini, which is the focus of this study, includes the morphologically distinctive genera *Equulites* and *Photolateralis* both of which exhibit lateral translucent windowing posterior to the pectoral region in males [30,34,35].

The genus *Equulites* includes 10 species distributed throughout the family's range, from the Red Sea and Madagascar to Japan and eastern Australia, and is in part diagnosed by having a slender or oblong body profile, a dorsal pigmentation pattern composed of vermiculations or oblong lines, a downward protruding mouth, and both a translucent lateral flank patch and a lateral translucent patch in the ventrolateral gas bladder and body wall in mature males [30,34]. Like the lateral windowing, the light organ in *Equulites* is sexually dimorphic, especially so in the slender ponyfish *E. elongatus* and the offshore ponyfish *E. rivulatus* where the adult male light organs are two or more times the size of light organs in similarly sized conspecific females [2,5,10,12,36]. Haneda and Tsuji [5] used gross dissection to describe the external and internal anatomy of the light organ in *E. elongatus* finding that the light organ in both male and female *E. elongatus* has left and right anterior lobes lateral to the esophagus and posterior lobes that contact the anterior gas bladder. Most of the internal area of the esophageal organ is composed of elongate, closely packed chambers containing bacteria and the organ has a ventral covering of dark pigment they called the "black membrane", with dark chromatophores, iridophores, and cells or cell clusters containing both dark pigment and guanine dorsal to the black membrane [5]. Additionally, they described dorsal left and right paired ducts connecting the lumen of the dorsal esophagus to left and right cavities that extend most of the length of the light organ that they suggest allows entry and egress of bacteria [5].

Observations of lateral bioluminescent blinking behavior in wild male *Equulites* suggests that blinking has a role in reproductive communication. Male *E. elongatus* in mixed male-female groups at night during spring and summer blinked approximately every one second, and in some observations, males coordinated their blinks. Observers documented bioluminescence laterally, and during these observations did not observe ventral counterillumination luminescence in either males or females [24]. The anatomical basis for blinking behavior most likely involves modifications of the light organ via expansion and contraction of dark chromatophores, the muscular movement of a screening sheet of tissue, or both [5]. The rapidity of the blinking observed strongly suggests a muscular mechanism for lateral blinking as was suggested by Haneda [22] when considering blinking reported in other genera of ponyfishes.

The genus *Photolateralis* includes four species distributed from Madagascar and the Gulf of Oman to Guam and Australia, in part diagnosed by having an oblong lateral profile, and lateral windowing composed of a continuous or composite midlateral stripe in mature males that laterally overlaps only the posterior region of the gas bladder [35,37]. The light organ is sexually dimorphic in *Photolateralis*, but this dimorphism is less pronounced than in *Equulites*, and the light organ exhibits less posterodorsal expansion [2,38]. The lateral translucent windowing of the body wall varies considerably in extent among the four species in this genus [37,39].

The tribe Equulitini presents an opportunity to explore the range of anatomical variation in the light organ in a well diagnosed, strongly supported clade of sexually dimorphic ponyfishes, with one member that has been documented to blink [12,24]. Although many studies generally described the external anatomy of the leiognathid light organ [10,12,15,38], the only study of internal structure involved gross dissection and study of luminescent behavior in live and newly dead specimens [5], and as a result the specific tissues composing the light organ have not been described. In this study we seek to explore light organ variation in the tribe Equulitini and to provide the first histological characterization of the light organ of leiognathids to identify the tissues associated with bioluminescent-organ functions. Recognizing that rapid blinking likely would require muscular involvement, we predict that the muscles of the esophagus will exhibit some modification to facilitate blinking.

## Materials and methods

### Specimens and gross examination

We examined, histologically sampled, and diffusible iodine-based contrast-enhanced computed tomography (diceCT) scanned preserved specimens in museum collections (Table 1). Throughout this paper institutional codes for ichthyological collections follow Sabaj [40,41], PVD indicates field numbers for unaccessioned collections by Paul V. Dunlap, and

**Table 1. Specimens examined.**

| Species | Museum Catalog Numbers |
|---|---|
| *Equulites elongatus* | AMNH uncat. PVD 82–06/19a*†, LACM 4307–1, 43612–1, SIO 96–98, USNM 383296 |
| *Equulites klunzingeri* | ANSP 83337 |
| *Equulites leuciscus* | AMNH 245614, uncat. PVD 00–10/18*†, uncat. PVD 02–01/13†, uncat. PVD 02–01/30r†, CAS 38786, USNM 191977, THAI 24–56 |
| *Equulites oblongus* | ANSP 8772 |
| *Equulites popei* | SIO 83−55 |
| *Equulites rivulatus* | AMNH uncat. PVD 82–06/19a*†, SIO 83–55, 96–98 |
| *Equulites* sp. | JFBM 48712* |
| *Leiognathus equulus* | AMNH 269767 |
| *Nuchequula nuchalis* | AMNH 269790 |
| *Photolateralis moretoniensis* | AM I.21700001†, AM I.22983001† |
| *Photolateralis stercorarius* | AMNH uncat. PVD 99–11/30a*, PVD 03–04/07a†, SIO 00−13, CAS 20004, 51104, SIO 96−96, ZRC 36659–36666, 41764 |
| *Photopectoralis aureus* | AMNH 241290, 89922, LACM 43590−1, SIO 83132, UMMZ 240129 |

*, histologically sampled; †, iodine stained μCT scanned.

THAI indicates unaccessioned specimens examined at Chulalongkorn University in Bangkok. We did not work directly with live animals and used dead specimens obtained from fish markets and/or deposited in a museum collection (Regis University Institutional Animal Care and Use Committee exempted, 2018 letter).

We examined ethanol-preserved museum specimens using a Leica MZ 12.5 dissecting stereomicroscope with an Amscope MU 10 MP camera at Regis University and a Nikon SMZ800 dissecting stereomicroscope outfitted with an Excelis 4K UHD camera at the American Museum of Natural History (AMNH) to examine and photograph specimens. We used halogen lighting via fiber-optic light guides for examination of specimens on the dissecting steromicroscope. We photographed fresh specimens using a Sony A7RV mirrorless camera with Sony 90 mm macro lens under ambient light. We used a Semrock (Rochester, NY) band pass filter 482 + /- 18 nm affixed to a LED flashlight for illumination within the spectrum of bacterially produced light in ponyfishes [5,6,8]. For internal examination, previous researchers dissected many of the examined specimens. When additional dissection was needed, we cut immediately right parasagittally along the ventral surface and then dorsally on the right side anterior to the anal fin to the dorsal margin of the visceral cavity. We then cut dorsally on the right side under the posterior operculum and through the pectoral girdle. We then reflected the body wall dorsally to expose features of the light-organ system. We placed particular emphasis on examining and documenting the structure of the esophageal light organ, the gas bladder, and surrounding tissues.

## Histological sectioning

We removed samples for histological study from ethanol preserved specimens by excising the esophagus, including the light organ. We dehydrated samples via an ethanol series to 100% ethanol, followed by a xylene series, and paraffin infiltration. We embedded samples in paraffin blocks and sectioned them every 10 µm on an American Optical Spencer 820 manual rotary microtome at Regis University and a Leica HistoCore MULTICUT rotary microtome at the AMNH. We mounted sections on charged glass slides using a water bath at 47°C [42], and stained every other slide with the Masson's trichrome (MT) protocol to differentiate collagen and muscle [43,44] and Toluidine Blue O pH 4.1-(TB) as a general stain to differentially stain cartilage and retain visible guanine crystals [45]. We cover slipped stained sections using a toluene-based synthetic resin mounting medium, then examined and photographed with a Leica DM 2500 compound microscope with an Amscope MU 10 MP camera at Regis University and a Nikon Eclipse 50i compound microscope with an Excelis 4K UHD camera at the AMNH. We prepared histological images for figures by increasing brightness and contrast evenly across the entire image and eliminating fragments or discoloration in the mounting medium outside the external margin of the tissue using GIMP, GNU Image Manipulation Program 3.0.4 [46].

## CT scanning

We used diceCT [47–49] to view soft tissues *in situ* in formalin-fixed specimens stored in 75% ethanol. We initially hydrated specimens via a decreasing percentage ethanol series followed by five to seven days in a 20% Lugol's Iodine solution (approximately 0.82% iodine and 1.25% potassium iodine solution) [48,49]. We scanned specimens at the AMNH using a Zeiss VersaXRM 730 high-resolution 3D X-ray microscope (µCT) with ZEN navx 2.0 Control System software set to a flat panel scan, resulting in a voxel resolution of between 8.5 and 16.0 µm. We first cropped TIFF image stacks in Fiji 1.54r, including ImageJ [50] and converted to nearly raw raster data format. We then used 3D Slicer 5.8.1 [51,52] to sample section images and for 3D segmentation and reconstruction. We processed images for use in figures using GIMP 3.0.4 [46] to crop and increase brightness and contrast evenly over the entire image when the iodine staining was less extensive. Gas bladder lumina filled with air are visible as solid black regions and gas bladder lumina containing water are visible as uniform gray regions. We colored any gray gas bladder lumina black for consistency and contrast. We used Inkscape 1.4 [53] for final figure construction.

## Results

### Gross anatomy of the light-organ system

We observed lateral translucent windows in adult male specimens of *Equulites* and *Photolateralis* as gaps in the lateral silvering of the flank, which appears quite reflective, likely due to the arrangement of crystalline guanine [54,55]. External translucent windowing is absent in *Equulites* females and is either absent or greatly reduced in female *Photolateralis* specimens. In preserved male specimens in *Equulites*, the lateral translucent windows extend from the pectoral fin base to near the posterior extent of the gas bladder. The degree of translucent windowing varies in prominence when based on direction of viewing in both fresh and preserved specimens, with most individuals appearing to lack windowing and have continuous reflective silvery coloration when viewed dorsolaterally or to have faint or limited windowing when viewed laterally (Figs 1A and 2A). However, the translucent windows are prominent when viewed ventrolaterally in all male specimens suggesting but not conclusively demonstrating that guanine is present in the area but that the orientation of the guanine crystals is the likely mechanism determining the direction of light emission via a lenticular arrangement

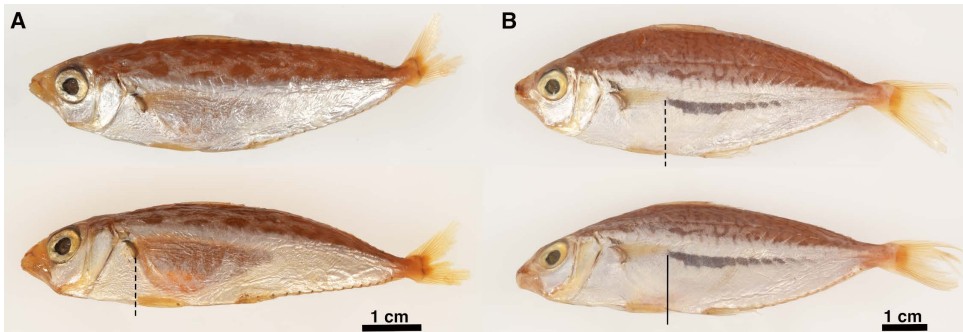

**Fig 1. Left lateral windowing in preserved adult males. (A)** *Equulites elongatus* (AMNH PVD 82-06/19a), 80.1 mm SL, top in lateral view, bottom in ventrolateral view. **(B)** *Photolateralis stercorarius* (AMNH PVD 99-11/30a), 97.2 mm SL, top in lateral view, bottom in ventrolateral view. Dashed line indicates anterior-most extent of visible translucent window. Photographs taken by J.S.S. and are original to this work.

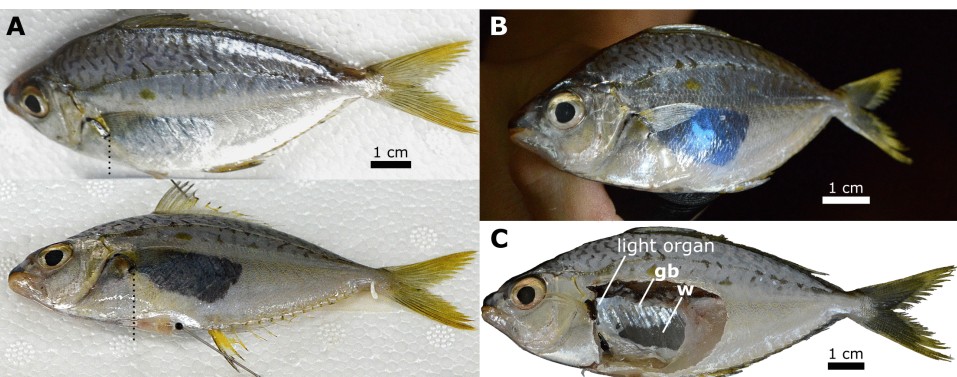

**Fig 2. Left lateral views of fresh adult male *Equulites leuciscus*.** THAI 24−56. Specimens on day of acquisition from fish market, before preservation. **(A)** Top in dorsolateral view and bottom in ventrolateral view under ambient light. Dashed line indicates anterior-most extent of visible window. **(B)** Lateral view of fish dissected on right side on right side and illumined with light at wavelength of most common bacterially produced light in ponyfishes (482 +/- 18 nm) through light-organ window into gas bladder (to approximate how light would travel, be reflected, and emitted in this species). **(C)** Lateral view of dissection under ambient light. Abbreviations: silvery reflective gas bladder **(gb)**, transparent gas bladder window **(w)**. Photographs taken by J.S.S. and are original to this work.

(Figs 1A and 2A–2C–2C). In preserved and fresh male specimens of *Photolateralis* the lateral translucent windows extend from over the posterior quarter or third of the gas bladder onto the anterior caudal region and are visible from all lateral points of view in both fresh and preserved specimens (Fig 1B). In a fresh *Equulites* specimen, placement of a light source in the position of the window into the gas bladder in a dissected male specimen of *E. leuciscus* resulted in clear emission of light via the translucent lateral flank window (Fig 2B).

The light organ is attached to the dorsal and lateral surfaces of the esophagus in all specimens examined and the dorsal lobe is located between the esophagus and the gas bladder (Figs 2C, 3, 4A and 4B). Lateral lobes remain separate

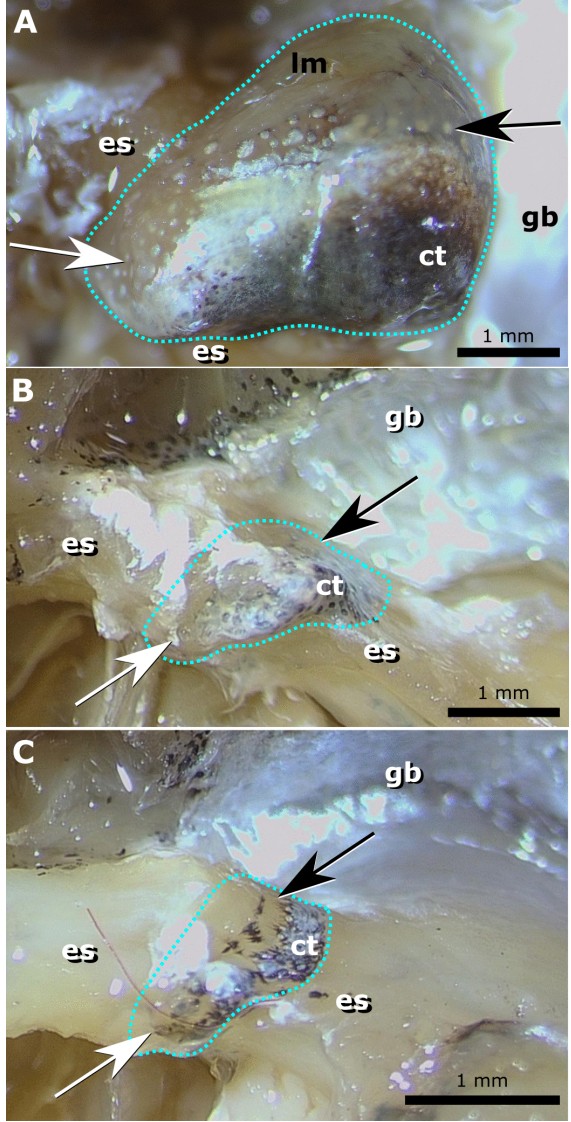

**Fig 3. Left lateral view of esophageal light organs *in situ*.** Light organ outlined in dotted line. **(A)** *Equulites rivulatus* male AMNH uncat. PVD 82-06/19a, 61.7 mm SL **(B)** *Photolateralis stercorarius* male AMNH uncat. PVD 03-04/07a, 62.0 mm SL. **(C)** *Photolateralis stercorarius* female AMNH uncat. PVD 03-04/07a, 62.5 mm SL. White and black arrows indicate position of anterior and posterior windowing respectively. Abbreviations: connective tissue with dark chromatophores and iridophores **(ct)**, esophagus **(es)**, gas bladder with reflective silvering **(gb)**, longitudinal striated muscle **(lm)**. Photographs taken by M.J.G. and are original to this work.

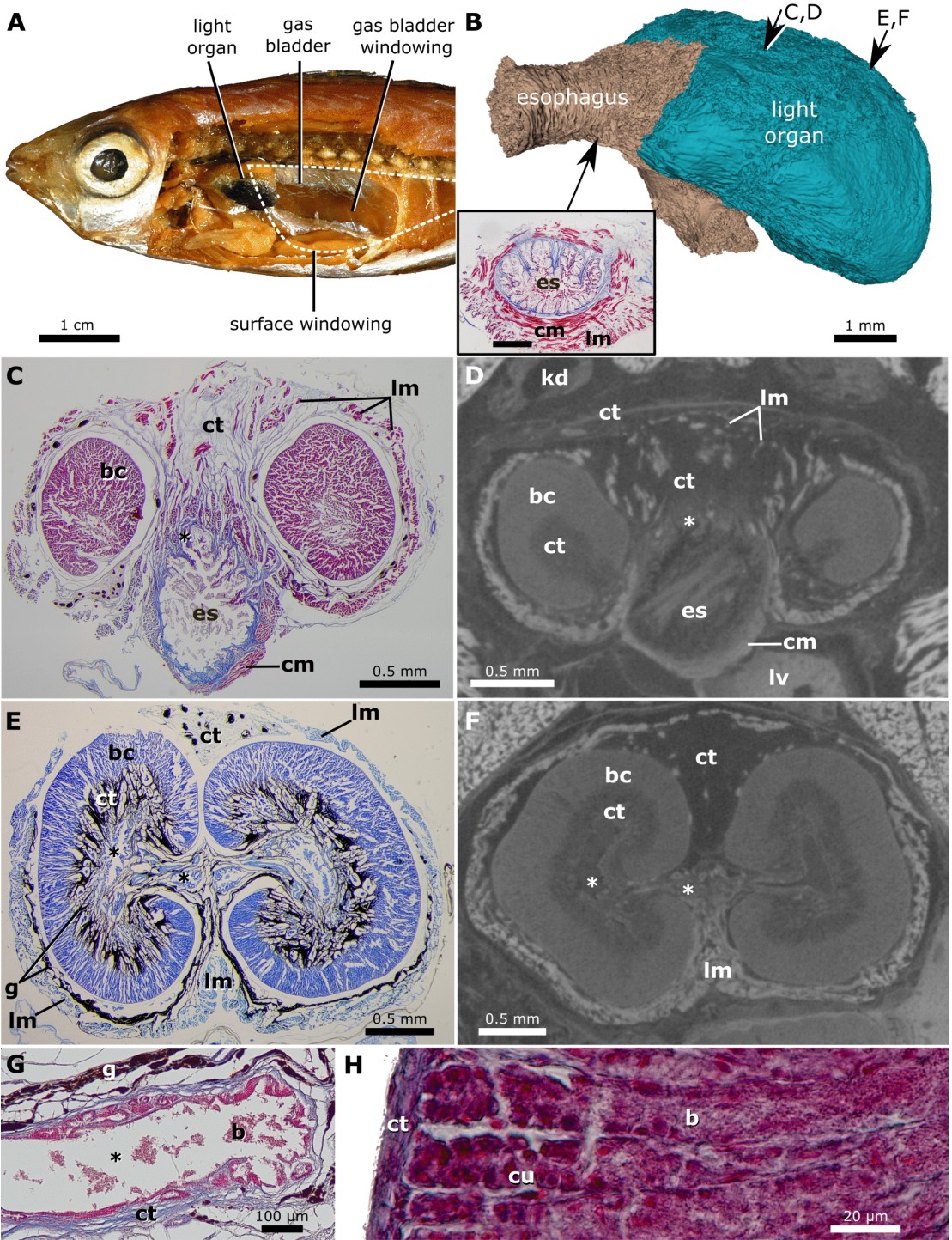

**Fig 4. Male *Equulites elongatus* (AMNH uncat. PVD 82-06/19a) light organ. (A)** Dissected specimen showing relative positions of light organ, gas bladder, and external windowing. Photograph taken by J.S.S. and is original to this work. **(B)** μCT reconstruction of light organ and esophagus in left anterodorsal view. Inset shows a section of the esophagus (MT stain). Scale bar, 0.5 mm. Arrows indicate approximate locations of cross sections. **(C)** Anterior cross section of light organ and esophagus showing connecting ducts (MT stain). **(D)** μCT cross section at approximate location of C. **(E)** Posterior cross section of light organ (TB stain). **(F)** μCT cross section at approximate location of E. **(G)** Cross section of central chamber from posterior region of light organ at a location similar to medial asterisks in E and F (MT stain). **(H)** Cross section of tubules from posterior light organ containing bacteria

(MT stain). Abbreviations: bacteria **(b)**, tubular bacterial chambers **(bc)**, circular striated muscle **(cm)**, connective tissue **(ct)**, simple cuboidal epithelium **(cu)**, ducts and chambers connecting to esophageal lumen **(*)**, esophagus **(es)**, guanine in connective tissue **(g)**, kidney **(kd)**, longitudinal striated muscle **(lm)**, liver **(lv)**.

anteriorly in species in the genus *Equulites* and contact each other along the ventral midline of the esophagus in *Photolateralis*. In the species examined outside Equulitini the light organ fully surrounds the esophagus ventrally and does not have obvious left and right lobes. The esophagus is longer and narrower in the more elongate species, e.g., *E. elongatus* and *P. stercorarius* and shorter and broader in the more deep-bodied species, e.g., *E. leuciscus* and *P. moretoniensis*, where it is as deep or deeper dorosoventrally as it is long anteroposteriorly. The dorsal component of the light organ is primarily in contact with the anterior surface of the gas bladder in examined specimens of *Equulites* (Figs 2, 3A and 4A), whereas the contact is ventral to the anterior gas bladder in *Photolateralis* and other examined leiognathid species (Fig 3B and 3C). At the point of contact there is an oval transparent window in the silvered lining of the gas bladder. The light-producing, bacterial component of the light organ is visible through this window as a light-colored area. This light-colored region is usually at least partially obscured from the ventral margin by a band with scattered dark chromatocytes and iridocytes with a region of continuous guanine and dark pigmentation further ventral to that and continuing along the ventral surface of the organ (Fig 3). The extent of the window that is obscured varies among individual preserved specimens. Some exhibit little screening of the bacterial component of the light organ, most exhibit some screening by the band of scattered cells, and some exhibit combined screening by the band of scattered cells and ventral solid screening tissue with continuous guanine and dark pigmentation.

The ventral surface of both the lateral lobes and the median dorsal component of the light organ are colored a continuous black and/or reflective silver with a visible band above the dark ventral area composed of tissue with scattered dark chromatocytes and iridocytes (Fig 5). This band is visible posteriorly as either a small, median dark area in the transparent window into the anterior gas bladder (Fig 5A) or a band of screening cells across the ventral or entire window (Fig 5B). A thin translucent layer of muscle fibers extends anterior to posterior along the dorsal surface of the organ, including the lateral lobes (Fig 5). The light-colored area containing the bacteria is visible through the dorsal and posterior longitudinal muscle (Fig 5B). Similar translucent longitudinal muscles extend anterior to posterior along the ventral light organ. The lighter dorsal and posterior areas are the surfaces of the light organ through which it is most likely that light would pass, as was assumed by prior authors [2,5,10,25].

## Histological anatomy of the light organ

The esophagus anterior to the light organ is lined by a shallowly stratified cuboidal epithelium with large numbers of mucus cells underlain by a mucosal connective tissue layer. Between the connective tissues of the mucosa and submucosa there are thin areas of longitudinal muscle that are irregularly scattered around the diameter of the organ and compose a diffuse muscularis mucosae that is absent in some sections. Because of the lack of consistent differentiation of mucosal and submucosal all the connective tissue between the epithelium and muscularis propria will be referred to as mucosal hereafter. The thick muscularis propria layer of the esophagus (called the muscularis hereafter) is composed of deep circular muscle and superficial longitudinal muscle (Fig 4B). The esophagus is dorsoventrally elongate between the lateral lobes of the light organ, and at the anterior-most portion of the dorsal light organ, the lumen of the esophagus is connected via multiple dorsal ducts with the ducts, chambers, and tubules within the light organ (Fig 4C). In the areas where the esophagus is surrounded by the light organ dorsally, the esophagus has a reduced muscularis and a thicker mucosal layer of connective tissue.

The musculature of the light organ in histologically examined equulitine species differs from the musculature of the anterior esophagus in having more prominent, extensive longitudinal musculature and less circular musculature (Fig 4C and 4E). In both males and females, the more superficial longitudinal muscles connect to flattened dorsal and ventral

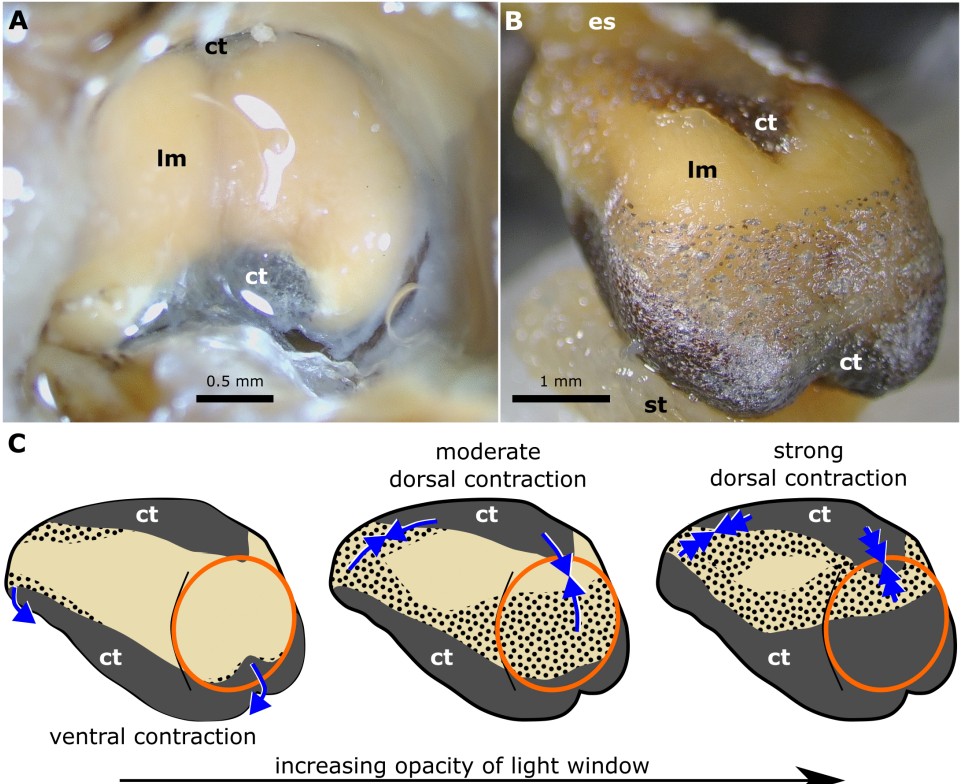

**Fig 5. Male *Equulites elongatus* light organ in posterior view. (A)** Light organ in posterior view from anterior gas bladder *in situ* (SIO 96−98, 83.1 mm SL) with screening tissues retracted. **(B)** Dorsoposterior view of excised light organ, esophagus, and anterior stomach (AMNH PVD 82-0/19a, 76.3 mm SL) with ventral screening band of tissue partially extended. **(C)** Diagrammatic light organs in left posterolateral view. Left image shows an unob-scured translucent window into the gas bladder similar to photo in (A). Middle image shows a translucent window with shutter partially covering window with tissue containing separated chromatocytes and iridocytes similar to photo in (B). Right image shows a translucent window with shutter covering window obscuring light into the gas bladder. Orange oval indicates window into gas bladder. Blue arrows indicate direction and degree of longitudinal muscle contraction. Abbreviations: connective tissue **(ct)**, esophagus **(es)**, longitudinal muscle **(lm)**, stomach **(st)**. Photographs taken by M.J.G. and are original to this work.

structures composed of connective tissue and muscle that contain dark chromatocytes and guanine and overlay the deeper structures of the light organ (Fig 5B, 5C and 6A). These muscular structures likely function as shutters and facili-tate rapid occlusion of the light-organ windows and prohibit bacterially generated luminescence from being emitted. The dorsal shutter contains circular muscle and is connected to longitudinal muscle that extends over the dorsal organ con-necting dorsal and ventral shutters (Fig 6A and 6B). When the dorsal longitudinal muscles contract this likely would pull the dorsal and ventral shutters together.

We observed an extensive system of internal ducts and cavities connecting the lumen of the esophagus to the densely packed light-organ tubes containing bioluminescent bacteria. The multiple ducts leading from the dorsal esophagus into the light organ are lined by a simple cuboidal epithelium that is continuous with the ciliated simple columnar epithelium lining the bilateral larger chambers that contain scattered groups of bacteria (Figs 4E, 4G, 6C and 6D). The left and right chambers connect to subdividing ducts that terminate in densely packed tubules containing dense populations of bacteria (Figs 4 and 6). The epithelium transitions from columnar to cuboidal along the subdividing ducts (Fig 6C) with the termini of the bacterial tubules having an especially prominent simple cuboidal epithelium (Fig 4H). The extensive connective tissue between the chambers and ducts contains substantial amounts of extracellular light-reflective guanine crystals that

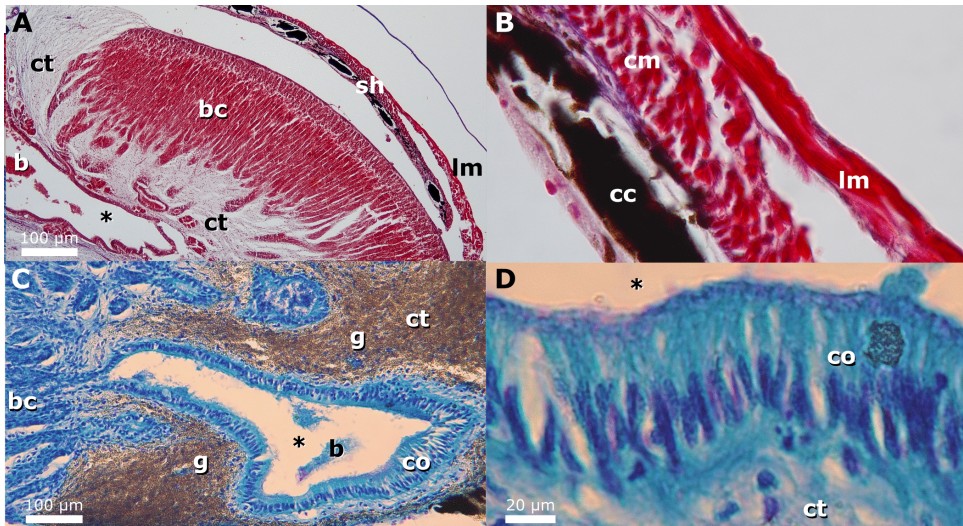

**Fig 6. Histological sections of *Equulites leuciscus* and *Photolateralis stercorarius* light organs. (A)** *Equulites leuciscus* male light organ longitudinal section showing the muscular, dorsal light occluding shutter (AMNH PVD 02-01/30r, 63.3 mm SL) (MT stain). **(B)** Close up view of longitudinal section of shutter from same section shown in A (MT stain). **(C)** Cross section of central chamber from posterior light organ connecting to tubular bacterial chambers at left in *Photolateralis stercorarius* (AMNH PVBD 03-04/07a, 57.7 mm SL) (TB stain). **(D)** Close up view of ciliated columnar epithelium lining central chamber in *P. stercorarius* (TB stain). Abbreviations: bacteria **(b)**, bacterial chambers **(bc)**, chromatocyte **(cc)**, circular striated muscle **(cm)**, columnar epithelium **(co)**, connective tissue **(ct)**, ducts and chambers connecting to the esophageal lumen **(*)**, guanine **(g)**, longitudinal striated muscle **(lm)**, shutter **(sh)**.

are visible in Toluidine blue stain preparations as gray to black elongate inclusions (Figs 4E and 6C). These are not visible in the Masson's trichrome preparation, wherein the guanine crystals dissolve during the acidic steps (e.g., Boin's solution, phosphotungstic acid) [44].

### CT anatomy of the light organ

Dice μCT scanning confirms the general morphology described in the histology section but provides a better understanding of the structure of the light organ *in situ* and in three dimensions (Fig 4B–4F). The three-dimensional reconstructions illustrate the presence of lateral lobes that do not contact each other along the ventral midline in the specimens of *Equulites* examined and do contact each other in the specimens of *Photolateralis* examined (Fig 7). Sectional views demonstrate that the lumen of the esophagus near the midline was notably as deep or deeper dorsoventrally than long anteroventrally in the deeper-bodied species (*E. leuciscus*, *E. rivulatus*, and *P. moretoniensis*), and that there are no identifiable extrinsic muscles inserting on light-organ structures. In members of the genus *Equulites*, the anterior gas bladder was principally situated posterior to the light-organ window (Fig 5A), whereas in both the species of *Photolateralis* and non-equulitine species examined, it was situated dorsal to the light-organ window (Fig. 7). The μCT scans also allowed for a more extensive survey to determine the direction of muscle fibers in the light organ, demonstrating a reduced proportion of circular musculature versus longitudinal muscle as compared to the condition in the anterior and posterior esophagus where it is not closely associated with the light organ (Fig 4D and 4F).

### Discussion

We provide the first detailed, tissue-level description of the light organ system in ponyfishes. Our observations are consistent with the prior broad-scale observations of external [3,10,12,15] and internal anatomy [5]. As a developmental outgrowth of the dorsal esophagus [7], the light organ's function is reflected in the modification of the esophageal layers when

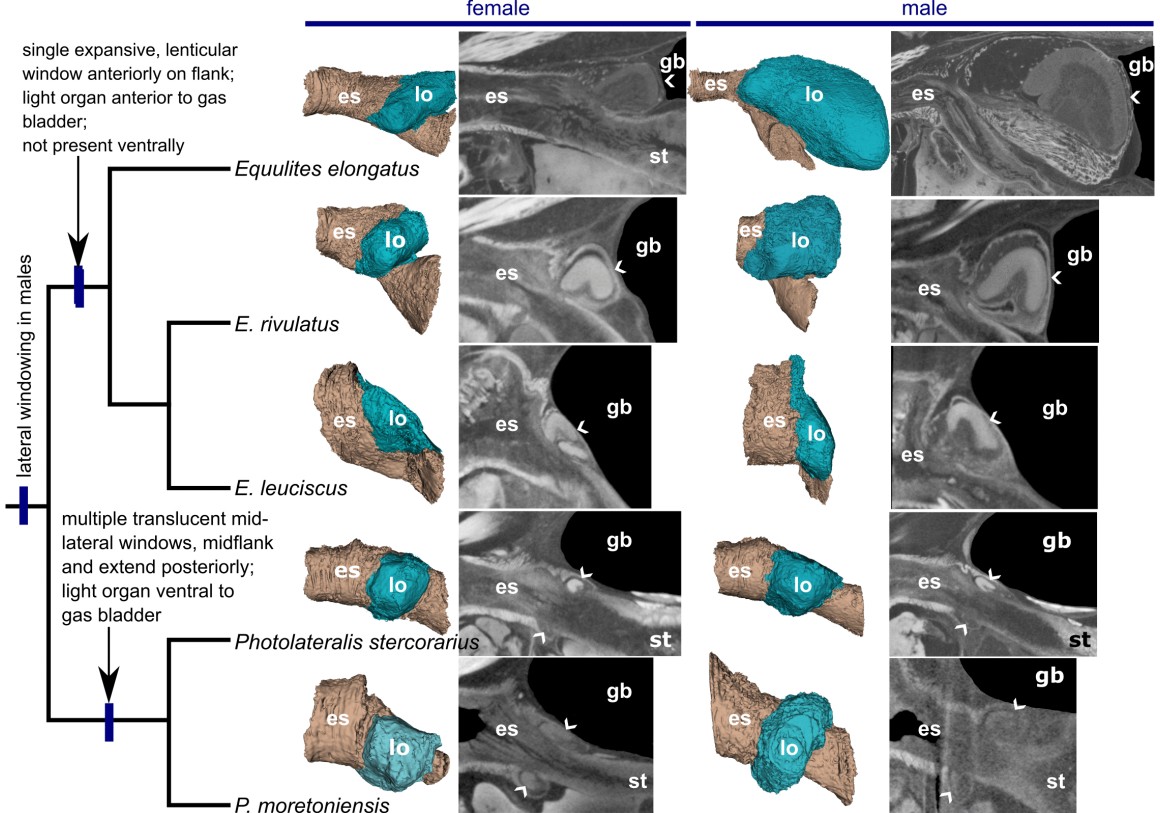

**Fig 7. Dice µCT scanned three-dimensional and sectional reconstructions of light organs in five equulitine species shown in phylogenetic context.** Phylogeny at left from the scientific literature [12,15,30]. Longitudinal CT sections are on or very near the median line. The chevrons indicate approximate locations of light windows in light organs along the median line. Abbreviations: esophagus **(es)**, gas bladder **(gb)**, light organ **(lo)**, stomach **(st)**.

compared to the typical esophagus structure visible anterior to the light organ (Fig 4B). The light organ contains symbiotic bioluminescent bacteria that are densely packed in the more superficial tubules and are more loosely distributed in the deeper ducts and chambers (Figs 4E, 4G, 4H and 6C). The ciliated columnar epithelium lining the large, paired chambers likely facilitates the movement of bacteria into and out of the tubules and is continuous with the esophageal lumen via multiple ducts (Fig 6C and 6D). This system connects to the esophageal lumen via two or more ducts on either side (Fig 4C and 4D), not via the single left and right ducts described in Haneda and Tsuji [5]. The extensive mucosal connective tissues contain visible extracellular guanine that likely reflects the light produced by the bacteria and helps to direct light out of the dorsal windowed parts of the light organ (Fig 6C). The muscularis also exhibits reduced musculature with proportionally much less circular muscle than longitudinal muscle as compared to the unmodified esophagus (Fig 4).

The externally visible ventral pigmentation and the histologically visible ventral and lateral presence of guanine in both the deep and superficial connective tissues (Fig 4E) allows light to exit the primarily dorsal windows into surrounding tissues and structures. In *Equulites* and *Photolateralis* the windowing on the posterior light organ into the gas bladder is directed either posteriorly to the gas bladder in *Equulites* or posterodorsally in *Photolateralis* (Fig 7). This means that in members of both genera the reflected light is transmitted to the posterior body, producing posteroventral counterillumination and illuminating lateral windows in the body wall of males (Figs 2 and 6). Anterior windowing occurs on the lateral lobes with the angle and extent of the lobes resulting in the windows being directed anterodorsally along the lateral body only in *Equulites* and anteriorly in *Photolateralis*. In *Photolateralis* this is consistent with the transmission of light through transparent tissues to produce anteroventral counterillumination (Fig 4). It is unclear if the dorsally emitted light from

windows on the anterior lobes in *Equulites* is subsequently reflected by other tissues anteroventrally for counterillumination. We did not observe any obvious pattern of additional reflective tissues in this orientation that would be indicative of anteroventral counterillumination. The intensity of light emission by *E. elongatus* was described as "feeble" by Haneda [22] who focused on counterillumination but lateral illumination in males was noted to be prominent by Sasaki et al. [24]. It may be that anterior counterillumination is either reduced or produced over less of this ventral surface in the genus *Equulites*.

Observations of rapid blinking in multiple species of ponyfishes [2,5,23,24] suggests that a muscular mechanism is obscuring and revealing the light produced by the symbiotic bacteria housed in the light organ, in a way that may be similar to what occurs in the bacterially bioluminescent flashlight fishes of the family Anomalopidae [56]. Previous authors provided first documentation of the presence of muscular dorsal and lateral shutters in species in other Tribes of ponyfishes, indicating that they are part of the wall of the light organ itself and assert that they obscure emitted light with opaque tissue [2,5] but these studies of live and gross anatomy do not identify the specific esophageal tissues that comprise these shutters. We document a shutter that likely is moved by longitudinal muscles during blinking in members of Equulitini. Contraction of dorsal longitudinal muscles would depress the dorsal shutter and substantially elevate the ventral shutter, obscuring light, whereas contraction of the ventral longitudinal muscles and relaxation of the dorsal longitudinal muscles likely opens the shutters emitting light (Figs 5, 6A and 6B). Thin circular muscles are present in the shutters themselves, likely providing structural integrity, but they do not insert on tissues that could function as additional lateral shutters (Fig 6A and 6B). The opposing dorsal and ventral longitudinal muscles of the light organ likely presumably can produce different levels of screening. Moderate contraction would cover the windows with the region of the shutter that contains scattered dark chromatophores and iridophores, which should allow for the relatively slow expansion and contraction of these cells to adjust the level of light emitted in a way that might best produce camouflaging counterillumination matched to ambient light (Fig 5B and 5C). Strong dorsal contraction of longitudinal muscles that pull the band of dark chromatophores and iridophores after complete cellular expansion and/or the ventral-most continuous guanine and pigment over the light-organ windows, possibly fully obscure the light released by the light organ (Fig 5C). Prior observations of both males and females blinking in some species of ponyfishes [2,5,23] is consistent with the presence of similar muscular and mucosal structures forming shutters in both males and females of the species we examined. However, as is the case with flashlight fishes [56], specifics of how blinking is achieved may vary among ponyfish species. Longitudinal muscle in the esophagus typically shortens the esophagus regionally during swallowing and peristalsis [57,58], a function analogous to its role in pulling the mucosal shutters dorsally and ventrally in the light organ, but such a change in function would;likely have required independent nervous control of the dorsal and ventral musculature.

The tribe Equulitini, as confirmed by our observations, is diagnosed by the possession of translucent lateral patches on the flank in males [12]. Our examination of specimens and the descriptions of light organs in the literature [3,12,22,19] support the recognition of two light-organ characteristics as unique to *Equulites* within Leiognathidae, lateral light-organ lobes that do not meet at the ventral midline of the esophagus and light organs that are primarily situated anterior to the gas bladder (Fig 7). Additionally, the lateral translucent patches in *Equulites* males begin immediately posterior to the pectoral fin, exhibit a lenticular transparency in both fresh and preserved specimens wherein the angle of guanine crystals determines from which angle bacterial light is visible (Figs 1 and 2), and are positioned over lateral translucent windows in the guanine-lined gas bladder in addition to the posterior and ventral gas-bladder windowing present in all leiognathids [30](Fig 4). The sister genus, *Photolateralis*, exhibits the common leiognathid characteristics of having lateral light-organ lobes that meet at the ventral midline of the esophagus and a light organ that is ventral to the anterior gas bladder [3,12,22,19]. The lateral translucent windows in male individuals of *Photolateralis* are situated more posteriorly than in *Equulites* with only the anterior-most part of the windows lateral to the posterior gas-bladder windowing that is present in all leiognathids, and with windowing that extends further posteriorly on the flank than in *Equulites* [30] (Fig 1).

The light organ of leiognathids developmentally and evolutionarily is derived from a diverticulum of the dorsal esophagus, immediately anterior to the pneumatic diverticulum that subsequently forms the gas bladder [7], and the symbiotic relationship with bioluminescent bacteria occurs early in development well before the light organ achieves its

adult complexity [7]. After the light organ has developed, it exhibits most of the structural layers present in the wall of the unmodified esophagus and maintains a connection to the esophageal lumen. This continuing connection allows for bacterial colony maintenance via continuing egress and entry of bacteria, that likely is facilitated by the ciliated columnar epithelium in the large left and right chambers of the light organ. Use of larval fish collections to more specifically explore esophageal tissue differentiation could in the future provide a better understanding of both nascent light organ formation and establishment of the bacterial-fish symbiotic relationship.

## Conclusion

In this study we provide the first detailed description of the internal structure of the bioluminescent light organ in Leiognathidae (Figs 4–7) and specifically explore the anatomical and histological structure of the light organ in five species within the ponyfish tribe Equulitini (Fig 6). The light organ's structure reflects its esophageal origin with its ducts, chambers, and bacteria containing tubules that connect to the esophageal lumen with a continuous epithelium of cuboidal or columnar cells. Further, the light organ exhibits a muscularis with less circular muscle and an extensive mucosa that contains the silvery reflective guanine crystals that either reflect or screen light (Fig 3). The orientation and lateral position of the lateral lobes of the light organ in *Equulites* may suggest that these species either generate their anteroventral counterillumination via a different reflective pathway or that they emit less counterillumination anteriorly. Our results reveal that the light-organ shutters previously mentioned in the literature for various leiognathid species [2,5], but never specifically identified or described anatomically, are composed of chromatocytes and iridocytes, and guanine-containing mucosal tissues attached to transparent longitudinal muscle that extends over the dorsal and ventral surfaces of the light organ of both males and females. This transparent longitudinal muscle likely contracts to pull the shutter-like screening tissues to facilitate blinking and adjustment of the emitted light level (Figs 5, 6A and 6B). The lenticular organization of guanine crystals in the lateral surface windows of male *Equulites* (Figs 1A and 2A) are unique to this genus and allow ventrolateral light emission and dorsal light occlusion. Considering the range of diversity within the family generally, future examination of additional species of leiognathids likely will reveal additional structures and specializations of the light-organ system within this group.

## Acknowledgments

D. Catania of CAS, P. Hastings and H.J. Walker of SIO, W. Ludt, J. Seigel & C. Thacker of LACM, M. McGrouther & Y.-K. Tea of AMS, D. Nelson of UMMZ, L. Parenti of USNM, M. Sabaj Pérez of ANSP, A. Simons & K. Ford of JFBM, and R. Thoni of AMNH loaned specimens in their care, supported specimen use, allowed histological sampling, and/or allowed CT scanning for this study. P. Fu and K. Hurdle, AMNH Microscopy and Imaging Facility, provided essential support for CT scanning. We are grateful to S. Panha, P. (Pomme) Tongkerd (Chulalongkorn University, Bangkok), and their students for facilitating our collections and studies in Thailand. We also thank J. Limpichat (Mahidol University, Kanchanaburi), D. Boyd (LSU), and R. Thoni (AMNH) for assistance with packing, curating, and shipping specimens.

## Author contributions

**Conceptualization:** Michael J. Ghedotti, Rene P. Martin, Emily M. Carr, John S. Sparks.

**Data curation:** Michael J. Ghedotti.

**Formal analysis:** Michael J. Ghedotti.

**Funding acquisition:** Michael J. Ghedotti, John S. Sparks.

**Investigation:** Michael J. Ghedotti, Jordon J. Valdez, John S. Sparks.

**Methodology:** Michael J. Ghedotti.

**Project administration:** Michael J. Ghedotti.

**Supervision:** Michael J. Ghedotti.

**Validation:** Rene P. Martin, Emily M. Carr, John S. Sparks.

**Writing – original draft:** Michael J. Ghedotti.

**Writing – review & editing:** Michael J. Ghedotti, Jordon J. Valdez, Rene P. Martin, Emily M. Carr, John S. Sparks.

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
