## [Decision Letter · Decision Letter 0]

22 Mar 2026

PONE-D-26-12952Morphology of the light-organ system and bioluminescent blinking in the ponyfish tribe Equulitini (Teleostei: Leiognathidae)PLOS One

Dear Dr. Ghedotti,

Thank you for submitting your manuscript to PLOS ONE. After careful consideration, we feel that it has merit but does not fully meet PLOS ONE’s publication criteria as it currently stands. Therefore, we invite you to submit a revised version of the manuscript that addresses the points raised during the review process.

If applicable, we recommend that you deposit your laboratory protocols in protocols.io to enhance the reproducibility of your results. Protocols.io assigns your protocol its own identifier (DOI) so that it can be cited independently in the future. For instructions see: https://journals.plos.org/plosone/s/submission-guidelines#loc-laboratory-protocols. Additionally, PLOS ONE offers an option for publishing peer-reviewed Lab Protocol articles, which describe protocols hosted on protocols.io. Read more information on sharing protocols at . Additionally, PLOS ONE offers an option for publishing peer-reviewed Lab Protocol articles, which describe protocols hosted on protocols.io. Read more information on sharing protocols at https://plos.org/protocols?utm_medium=editorial-email&utm_source=authorletters&utm_campaign=protocols..

We look forward to receiving your revised manuscript.

Kind regards,

Caio Santos Nogueira, PhD

Academic Editor

PLOS One

Journal Requirements:

“Funding for this work was provided by the American Museum of Natural History to J.S.S. and E.M.C. as general departmental funding and a Regis University URSC Faculty Grant (URSCMJG2025) to M.J.G. https://one.regis.edu/academics/research-grants/ursc . The funders played no role in the study design, data collection and analysis, decision to publish, or preparation of the manuscript.”

We note that one or more of the authors is affiliated with the funding organization, indicating the funder may have had some role in the design, data collection, analysis or preparation of your manuscript for publication; in other words, the funder played an indirect role through the participation of the co-authors. If the funding organization did not play a role in the study design, data collection and analysis, decision to publish, or preparation of the manuscript and only provided financial support in the form of authors' salaries and/or research materials, please do the following:

a. Review your statements relating to the author contributions, and ensure you have specifically and accurately indicated the role(s) that these authors had in your study. These amendments should be made in the online form.

b. Confirm in your cover letter that you agree with the following statement, and we will change the online submission form on your behalf:

“The funder provided support in the form of salaries for authors [insert relevant initials], but did not have any additional role in the study design, data collection and analysis, decision to publish, or preparation of the manuscript. The specific roles of these authors are articulated in the ‘author contributions’ section.

“The American Museum of Natural History and Regis University provided facilities and equipment in support of this work. This research was funded, in part, by a Regis URSC Grant to M. Ghedotti. D. Catania of CAS, P. Hastings and H.J. Walker of SIO, W. Ludt , J. Seigel & C. Thacker of of LACM, M. McGrouther & Y.-K. Tea of AMS, D. Nelson of UMMZ, L. Parenti of USNM, M. Sabaj Pérez of ANSP,  A. Simons & K. Ford of JFBM, and R. Thoni of AMNH loaned specimens in their care, supported specimen use, allowed histological sampling, and/or allowed CT scanning for this study. P. Fu and K. Hurdle, AMNH Microscopy and Imaging Facility, provided essential support for CT scanning.

We are grateful to S. Panha, Piyoros (Pomme) Tongkerd (Chulalongkorn University, Bangkok), and their students for facilitating our collections and studies in Thailand. We also thank Jirasin Limpichat (Mahidol University, Kanchanaburi), D. Boyd (LSU), and R. Thoni (AMNH) for assistance with packing, curating, and shipping specimens.”

“Funding for this work was provided by the American Museum of Natural History to J.S.S. and E.M.C. as general departmental funding and a Regis University URSC Faculty Grant (URSCMJG2025) to M.J.G. https://one.regis.edu/academics/research-grants/ursc . The funders played no role in the study design, data collection and analysis, decision to publish, or preparation of the manuscript.”

4. Please note that your Data Availability Statement is currently missing the repository name, DOI/accession number of each dataset or a direct link to access each database. If your manuscript is accepted for publication, you will be asked to provide these details on a very short timeline. We therefore suggest that you provide this information now, though we will not hold up the peer review process if you are unable.

5. We note that Figures 1, 2 and 4 in your submission contain copyrighted images. All PLOS content is published under the Creative Commons Attribution License (CC BY 4.0), which means that the manuscript, images, and Supporting Information files will be freely available online, and any third party is permitted to access, download, copy, distribute, and use these materials in any way, even commercially, with proper attribution. For more information, see our copyright guidelines: http://journals.plos.org/plosone/s/licenses-and-copyright.

a. You may seek permission from the original copyright holder of Figures 1, 2 and 4 to publish the content specifically under the CC BY 4.0 license.

Reviewers' comments:

Reviewer's Responses to Questions

**Comments to the Author**

1. Is the manuscript technically sound, and do the data support the conclusions?

Reviewer #1: Partly

Reviewer #2: Yes

2. Has the statistical analysis been performed appropriately and rigorously? 

Reviewer #1: N/A

Reviewer #2: N/A

3. Have the authors made all data underlying the findings in their manuscript fully available?

Reviewer #1: Yes

Reviewer #2: Yes

4. Is the manuscript presented in an intelligible fashion and written in standard English?

Reviewer #1: Yes

Reviewer #2: Yes

5. Review Comments to the Author

Reviewer #1: Abstract

General comments:

- Lines 19 - 23: I recommend designating “communication”, namely intra- or/and inter-specific communication. In addition to these two categories. consider including the ecological role of illumination (of surroundings) as well.

- Lines 31 – 34: Please take caution when stating “novel mechanism for blinking”, since producing flashes via regulation of the chromatophore-filled tissues “shutters” was previously described for ponyfishes in, for example, Haneda (1949).

Reference mentioned: Haneda Y .1950. Luminous organs of fish which emit light indirectly. Pacific Science. 4:214–227.

Introduction

General comments: Please avoid the term “chromatiridophore” in this section and posteriorly in the manuscript, since it is not a recognized category of chromatophore cells. If the authors are reporting a new type of chromatophore, please include in the manuscript the supportive results, as electronic microscopy (cellular arrangement and ultrastructure), immunolabelling of melanin and guanine (pigment identification in histological structures), and Fourier-transformed infrared spectroscopy (FTIR) and/or x-ray diffraction (XRD) (chemical identification). In the case that the authors lack the access to biological samples, equipment or technical infrastructures to perform an accurate report of a new chromatophore type, please provide support information in the manuscript of the properties of the colored structures (e.g., morphological, cellular, behavior to focused light) and describe the structures by using terminology such as “chromatiridophore-like” or “potential chromate-iridophore”. Since these structures are most likely a combination of pigmentary and structural color cells (in this case, melanophores lying above iridophores), similar to lizards (Saenko et al., 2013) and other vertebrates (Shawkey and D’Alba 2017), I recommend using an adequate and accurate terminology such as “combination of iridophore and chromatophore”.

References mentioned:

Saenko S, Teyssier J, Marel D, Milinkovitch MC. 2013. Precise colocalization of interacting structural and pigmentary elements generates extensive color pattern in Phelsuma lizards. BMC Biology. 11:105.

Shawkey MD, D’Alba L. 2017. Interactions between colour-producing mechanisms and their effects on the integumentary colour palette. Philosophical Transactions of the Royal Society B: Biological Sciences. 372:20160536.

- Lines 48 – 50: Same consideration as for Abstract lines 19 – 23.

- Lines 56 – 58: The bibliographic reference 13 details the reflective system of photophores in hatchetfishes. Please rephrase this sentence or replace this reference with another clearly supporting the information mentioned in this phrase.

Materials and Methods

- Lines 155 – 156: Consider mentioning the full designation for the abbreviation “diceCT”.

- Lines 159 – 168: To enhance the clarity for the readers, I suggest presenting in table format the information regarding the specimens used in this study.

- Lines 178 – 179: I recommend describing which type of illumination source (e.g., broad visible-spectra white lights such as tungsten, LED, halogen) was used to observe the reflection and coloration of the chromatophore-filled tissues.

Results

General comments: Consider more caution on the interpretation of morphological characteristics on the anatomical structures and their optical and light modulation properties without supportive data on these properties.

- Lines 224 – 227: Angle-dependent reflection is characteristic of iridophores, and such observation on the tissue suggests that the guanine iridophores have a degree of arrangement to reflect light from and to certain directions (see, for example, Denton and Land,1971; Land 1972). The suggestion that “ the guanine crystals determine the direction of light emission” would require the support of results, at least, regarding the iridophores angular arrangement in the reflective tissue, and angle between the light source and the observation/photograph. Please rephrase this sentence and take care of the interpretation for the observed reflective properties on the external tissues.

Mentioned references:

Denton EJ, Land MF. 1971. Mechanism of reflection in silvery layers of fish and cephalopods. Proceedings of the Royal Society B: Biological Sciences. 178:43-61.

Land MF. 1972. The physics and biology of animal reflectors. Progress of Biophysical and Molecular Biology. 24:75-106.

- Lines 245 – 247: I recommend adding morphological measurements for the length and thickness of the esophagus in the mentioned species.

- Lines 289 – 291: Please rephrase this sentence in a more cautious way when mentioning “through which light would be able to pass”, since this study does not contain or mentions any data on the light transmission properties (mainly spectra and intensity) of this tissue.

Discussion

General comments: Same considerations as for the Results section.

- Lines 379 – 391; 409 – 410; 417 – 422; 435 – 439: Please rephrase this sentence accordingly to the previously commented for lines 224 – 227 and 289 – 291 of Results section.

Conclusion

No comments on this section.

References

No comments on this section.

Figures and Tables

- Figures 1 and 2. Consider indicating the whole area of the LOS windows. These structures are difficult to distinguish in fixed and fresh specimens, so representing the anterior-most extension may not be clear to the readers.

- Figure 5. Please confirm all the abbreviations representing the morphological structures in the figure match the ones in the legend.

-Figure 7. The words in bold are difficult to distinguish for the reader. I suggest adjusting it by enhancing the lettering size or use normal text instead of bold.

Reviewer #2: Comments to authors

PONE-D-26-12952 “Morphology of the light-organ system and bioluminescent blinking in the ponyfish tribe Equulitini (teleostei: Leiognathidae). By Ghedotti et al. This manuscript explains the blinking mechanism of luminescence in ponyfish for the first time. The scientific importance is undoubted. However, for accepting this manuscript, I wish you would please give clear answers to the following questions.

Fig. 2B. The method of photographing this picture should be shown in the Materials and Methods section. Which light source did you use? At which angle is the light irradiated? The interpretation of this picture should also be discussed in the text. Is this specimen an ethanol-fixed specimen? I think ethanol-preserved specimens lose the transparency of the muscle.

Fig. 4. Why is ‘cm’ labeled as striated muscle? It might confuse the readers.

Fig. 5. Where is ‘bc’ located? Is the ‘screening tissue’ a part of the connective tissue?

Fig. 6. Which panels are for Photolaterlis stercorarius? It should be stated clearly in the legend.

It is stated that ‘fully obscure light release’ (Line 418), but in Fig. 5B, guanine particles on the screening tissue look very sparse. I am wondering if it can ‘fully’ obscure the light release?

The authors suggested the similarity between the shutter system of ponyfish and that of Anomalopidae. However, as shown in reference 56, the shutter mechanisms vary among the species of Anomalopidae. This means that the shutter system of Equulites shown in this study might not be able generalizable to all blinking ponyfish species.

The mechanism is complicated, and I recommend the authors include a conclusive schematic drawing to explain the mechanism for blinking (shutter movement).

6. PLOS authors have the option to publish the peer review history of their article (what does this mean?). If published, this will include your full peer review and any attached files.). If published, this will include your full peer review and any attached files.

.

Reviewer #1: **Yes:** José Rui Lima PaitioJosé Rui Lima Paitio

Reviewer #2: **Yes:** Yuichi ObaYuichi Oba

---

## [Author Response · Author response to Decision Letter 1]

1 Apr 2026

Also see cover letter.

Dear PLOS One Editors,

Thank you for your review and consideration of the manuscript PONE-D-26-12952 “Morphology of the light-organ system and bioluminescent blinking in the ponyfish tribe Equulitini (Teleostei: Leiognathidae)”. Based on the decision of acceptance after revision and in consideration of those revisions, we are submitting a revision of the manuscript that addresses all reviewer comments. We greatly appreciated the improvement of the manuscript due to the reviewers’ comments. Both a track changes and a “clean” version of the Text also is submitted. See below my electronic signature for a comment-by-comment explanation of how we addressed each identified revision.

Author Agreement: All authors agreed to the submission of this revised Research Article and have no conflicts of interest.

Significance: This manuscript is significant in its first description of the internal structure and anatomical components of the bioluminescent organ of ponyfishes (Leiognathidae). It also describes the modification of existing esophageal tissues to form a complex light organ in the Tribe Equulitini. Notably, it describes the structure that allows blinking in this bacterially bioluminescent group. It additionally discusses the characteristics associated with bioluminescence that are unique to the genera within Equulitini.

Modified Funding Statement for Online Submission Form: The funders Regis University and the American Museum of Natural History provided support in the form of salaries for authors M.J.G., E.M.C., J.S.S., and a Regis University URSC Faculty Grant (URSCMJG2025) to M.J.G. https://one.regis.edu/academics/research-grants/ursc provided funds for supplies. The funders played no role in the study design, data collection and analysis, decision to publish, or preparation of the manuscript. The specific roles of these authors are articulated in the ‘author contributions’ section.

All authors agree with this statement.

Data Availability: All specimens used in this study are available from their indicated museums and histological slides are deposited at the AMNH and μCT scan data are available at Zenodo (DOI: 10.5281/zenodo.19295041).

Thank you for consideration of our revised manuscript.

Sincerely,

Michael J. Ghedotti

Michael J. Ghedotti, Ph.D. <')}}}}}>< (he, him, his)

Professor of Biology | Regis College

3333 Regis Blvd., Denver, CO 80221 D-8

P 303.458.4091 | E mghedott@regis.edu | REGIS.EDU

These are the comments provided by the reviewers and by PLOS One. We outline how we addressed each below.

Journal comments.

1. Meet PLOS One style requirements

2. Thank you for stating the following financial disclosure: “Funding for this work was provided by the American Museum of Natural History to J.S.S. and E.M.C. as general departmental funding and a Regis University URSC Faculty Grant (URSCMJG2025) to M.J.G. https://one.regis.edu/academics/research-grants/ursc . The funders played no role in the study design, data collection and analysis, decision to publish, or preparation of the manuscript.” We note that one or more of the authors is affiliated with the funding organization, indicating the funder may have had some role in the design, data collection, analysis or preparation of your manuscript for publication; in other words, the funder played an indirect role through the participation of the co-authors. If the funding organization did not play a role in the study design, data collection and analysis, decision to publish, or preparation of the manuscript and only provided financial support in the form of authors' salaries and/or research materials, please do the following: Yes this is a correct statement of the support provided.

a. Review your statements relating to the author contributions, and ensure you have specifically and accurately indicated the role(s) that these authors had in your study. These amendments should be made in the online form. These are correct.

b. Confirm in your cover letter that you agree with the following statement, and we will change the online submission form on your behalf: “The funder provided support in the form of salaries for authors [insert relevant initials], but did not have any additional role in the study design, data collection and analysis, decision to publish, or preparation of the manuscript. The specific roles of these authors are articulated in the ‘author contributions’ section.” Done.

3. Thank you for stating the following in the Acknowledgments Section of your manuscript: “The American Museum of Natural History and Regis University provided facilities and equipment in support of this work. This research was funded, in part, by a Regis URSC Grant to M. Ghedotti. D. Catania of CAS, P. Hastings and H.J. Walker of SIO, W. Ludt , J. Seigel & C. Thacker of of LACM, M. McGrouther & Y.-K. Tea of AMS, D. Nelson of UMMZ, L. Parenti of USNM, M. Sabaj Pérez of ANSP, A. Simons & K. Ford of JFBM, and R. Thoni of AMNH loaned specimens in their care, supported specimen use, allowed histological sampling, and/or allowed CT scanning for this study. P. Fu and K. Hurdle, AMNH Microscopy and Imaging Facility, provided essential support for CT scanning. We are grateful to S. Panha, Piyoros (Pomme) Tongkerd (Chulalongkorn University, Bangkok), and their students for facilitating our collections and studies in Thailand. We also thank Jirasin Limpichat (Mahidol University, Kanchanaburi), D. Boyd (LSU), and R. Thoni (AMNH) for assistance with packing, curating, and shipping specimens.” We note that you have provided funding information that is not currently declared in your Funding Statement. However, funding information should not appear in the Acknowledgments section or other areas of your manuscript. We will only publish funding information present in the Funding Statement section of the online submission form. Please remove any funding-related text from the manuscript and let us know how you would like to update your Funding Statement. Currently, your Funding Statement reads as follows: “Funding for this work was provided by the American Museum of Natural History to J.S.S. and E.M.C. as general departmental funding and a Regis University URSC Faculty Grant (URSCMJG2025) to M.J.G. https://one.regis.edu/academics/research-grants/ursc. The funders played no role in the study design, data collection and analysis, decision to publish, or preparation of the manuscript.” Please include your amended statements within your cover letter; we will change the online submission form on your behalf. Done in both manuscript and cover letter.

4. Please note that your Data Availability Statement is currently missing the repository name, DOI/accession number of each dataset or a direct link to access each database. If your manuscript is accepted for publication, you will be asked to provide these details on a very short timeline. We therefore suggest that you provide this information now, though we will not hold up the peer review process if you are unable. Done via Zenodo (DOI: 10.5281/zenodo.19295041).

5. We note that Figures 1, 2 and 4 in your submission contain copyrighted images. All PLOS content is published under the Creative Commons Attribution License (CC BY 4.0), which means that the manuscript, images, and Supporting Information files will be freely available online, and any third party is permitted to access, download, copy, distribute, and use these materials in any way, even commercially, with proper attribution. For more information, see our copyright guidelines: http://journals.plos.org/plosone/s/licenses-and-copyright.

a. You may seek permission from the original copyright holder of Figures 1, 2 and 4 to publish the content specifically under the CC BY 4.0 license. We recommend that you contact the original copyright holder with the Content Permission Form (http://journals.plos.org/plosone/s/file?id=7c09/content-permission-form.pdf) and the following text: “I request permission for the open-access journal PLOS ONE to publish XXX under the Creative Commons Attribution License (CCAL) CC BY 4.0 (http://creativecommons.org/licenses/by/4.0/). Please be aware that this license allows unrestricted use and distribution, even commercially, by third parties. Please reply and provide explicit written permission to publish XXX under a CC BY license and complete the attached form.” Please upload the completed Content Permission Form or other proof of granted permissions as an "Other" file with your submission. In the figure caption of the copyrighted figure, please include the following text: “Reprinted from [ref] under a CC BY license, with permission from [name of publisher], original copyright [original copyright year].” Done. No images came from another publication or copyrighted source. The photographs all were taken by authors on the manuscript from original source material (specimens) specifically and solely for use in this manuscript. They have not otherwise appeared in the public sphere. The author taking each photograph was added to the relevant figure legends with a modified version of the required statement (because they were not re-printed) stating their originality in this manuscript. The two authors who took the photographs statements giving explicit written permission to publish all submitted images under a CC BY license and those were submitted as “Other” Documents.

6. If the reviewer comments include a recommendation to cite specific previously published works, please review and evaluate these publications to determine whether they are relevant and should be cited. There are no such recommendations. A reviewer cited literature to make points in the provided comments and did not suggest explicit citation of these references.

7. Please review your reference list to ensure that it is complete and correct. Done.

Reviewer #1:

Abstract

Lines 19 - 23: I recommend designating “communication”, namely intra- or/and inter-specific communication. In addition to these two categories. consider including the ecological role of illumination (of surroundings) as well. Done.

Lines 31 – 34: Please take caution when stating “novel mechanism for blinking”, since producing flashes via regulation of the chromatophore-filled tissues “shutters” was previously described for ponyfishes in, for example, Haneda (1949). Reference mentioned: Haneda Y .1950. Luminous organs of fish which emit light indirectly. Pacific Science. 4:214–227. Done. We stated that these things were generally described previously. We obviously cannot add the citation in the abstract, but cite these authors throughout the manuscript.

Introduction

General comments:

Please avoid the term “chromatiridophore” in this section and posteriorly in the manuscript, since it is not a recognized category of chromatophore cells. If the authors are reporting a new type of chromatophore, please include in the manuscript the supportive results, as electronic microscopy (cellular arrangement and ultrastructure), immunolabelling of melanin and guanine (pigment identification in histological structures), and Fourier-transformed infrared spectroscopy (FTIR) and/or x-ray diffraction (XRD) (chemical identification). In the case that the authors lack the access to biological samples, equipment or technical infrastructures to perform an accurate report of a new chromatophore type, please provide support information in the manuscript of the properties of the colored structures (e.g., morphological, cellular, behavior to focused light) and describe the structures by using terminology such as “chromatiridophore-like” or “potential chromate-iridophore”. Since these structures are most likely a combination of pigmentary and structural color cells (in this case, melanophores lying above iridophores), similar to lizards (Saenko et al., 2013) and other vertebrates (Shawkey and D’Alba 2017), I recommend using an adequate and accurate terminology such as “combination of iridophore and chromatophore”. Done. We removed “chromatiridophore” and “chromatiridocyte” throughout and just mention the combined pigment and guanine centers discussed by Harms (1928)[15] and Haneda & Tsuji (1976)[16].

References mentioned: Saenko S, Teyssier J, Marel D, Milinkovitch MC. 2013. Precise colocalization of interacting structural and pigmentary elements generates extensive color pattern in Phelsuma lizards. BMC Biology. 11:105. Used by reviewer as an example and explanation, no need to add citation to this manuscript.

Shawkey MD, D’Alba L. 2017. Interactions between colour-producing mechanisms and their effects on the integumentary colour palette. Philosophical Transactions of the Royal Society B: Biological Sciences. 372:20160536. Used by reviewer as an example and explanation, no need to add citation to this manuscript.

- Lines 48 – 50: Same consideration as for Abstract lines 19 – 23. Done.

- Lines 56 – 58: The bibliographic reference 13 details the reflective system of photophores in hatchetfishes. Please rephrase this sentence or replace this reference with another clearly supporting the information mentioned in this phrase. Done. Mentioned that this is not just a phenomenon in ponyfishes.

Materials and Methods

- Lines 155 – 156: Consider mentioning the full designation for the abbreviation “diceCT”. Done.

- Lines 159 – 168: To enhance the clarity for the readers, I suggest presenting in table format the information regarding the specimens used in this study. Done.

- Lines 178 – 179: I recommend describing which type of illumination source (e.g., broad visible-spectra white lights such as tungsten, LED, halogen) was used to observe the reflection and coloration of the chromatophore-filled tissues. Done.

Results

General comments: Consider more caution on the interpretation of morphological characteristics on the anatomical structures and their optical and light modulation properties without supportive data on these properties. Done.

- Lines 224 – 227: Angle-dependent reflection is characteristic of iridophores, and such observation on the tissue suggests that the guanine iridophores have a degree of arrangement to reflect light from and to certain directions (see, for example, Denton and Land,1971; Land 1972). The suggestion that “ the guanine crystals determine the direction of light emission” would require the support of results, at least, regarding the iridophores angular arrangement in the reflective tissue, and angle between the light source and the observation/photograph. Please rephrase this sentence and take care of the interpretation for the observed reflective properties on the external tissues. Done.

Mentioned references:

Denton EJ, Land MF. 1971. Mechanism of reflection in silvery layers of fish and cephalopods. Proceedings of the Royal Society B: Biological Sciences. 178:43-61. Used by reviewer as an example and explanation, no need to add citation to this manuscript.

Land MF. 1972. The physics and biology of animal reflectors. Progress of Biophysical and Molecular Biology. 24:75-106. Used by reviewer as an example and explanation, no need to add citation to this manuscript.

- Lines 245 – 247: I recommend adding morphological measurements for the length and thickness of the esophagus in the mentioned species. The need for something more quantitative is well taken, and we modified the manuscript to address this. We added relative statements about length and width. We did not compile data because this would require substantial scaling given the different general sizes of the fish and because we were trying to make a general obvious point.

- Lines 289 – 291: Please rephrase this sentence in a more cautious way when mentioning “through which light would be able to pass”, since this study does not contain or mentions any data on the light transmission properties (mainly spectra and intensity) of this tissue. Done.

Discussion

General comments: Same considerations as for the Results section.

- Lines 379 – 391; 409 – 410; 417 – 422; 435 – 439: Please rephrase this sentence accordingly to the previously commented for lines 224 – 227 and 289 – 291 of Results secti

---

## [Decision Letter · Decision Letter 1]

6 Apr 2026

Morphology of the light-organ system and bioluminescent blinking in the ponyfish tribe Equulitini (Teleostei: Leiognathidae)

PONE-D-26-12952R1

Dear Dr. Ghedotti,

We’re pleased to inform you that your manuscript has been judged scientifically suitable for publication and will be formally accepted for publication once it meets all outstanding technical requirements.

An invoice will be generated when your article is formally accepted. Please note, if your institution has a publishing partnership with PLOS and your article meets the relevant criteria, all or part of your publication costs will be covered. Please make sure your user information is up-to-date by logging into Editorial Manager at Editorial Manager® and clicking the ‘Update My Information' link at the top of the page. For questions related to billing, please contact  and clicking the ‘Update My Information' link at the top of the page. For questions related to billing, please contact billing support..

Kind regards,

Caio Santos Nogueira, PhD

Academic Editor

PLOS One

Additional Editor Comments (optional):

Dear Authors,

I am pleased to inform you that both reviewers have evaluated the second version of the manuscript and recommended its acceptance.

I congratulate you on this achievement and thank you for choosing PLOS ONE for the publication of your article.

I wish you success with the publication of your work.

Reviewers' comments:

Reviewer's Responses to Questions

**Comments to the Author**

1. If the authors have adequately addressed your comments raised in a previous round of review and you feel that this manuscript is now acceptable for publication, you may indicate that here to bypass the “Comments to the Author” section, enter your conflict of interest statement in the “Confidential to Editor” section, and submit your "Accept" recommendation.

Reviewer #1: All comments have been addressed

Reviewer #2: All comments have been addressed

2. Is the manuscript technically sound, and do the data support the conclusions?

Reviewer #1: Yes

Reviewer #2: Yes

3. Has the statistical analysis been performed appropriately and rigorously? 

Reviewer #1: N/A

Reviewer #2: N/A

4. Have the authors made all data underlying the findings in their manuscript fully available?

Reviewer #1: Yes

Reviewer #2: Yes

5. Is the manuscript presented in an intelligible fashion and written in standard English?

Reviewer #1: Yes

Reviewer #2: Yes

6. Review Comments to the Author

Reviewer #1: The manuscript alterations met the comments to the version previously submitted.

I recommend the publication of the manuscript, and I would like to congratulate the authors for such a nicely done and interesting work, noteworthy for the research on bioluminescence in teleosts.

Reviewer #2: (No Response)

7. PLOS authors have the option to publish the peer review history of their article (what does this mean?). If published, this will include your full peer review and any attached files.). If published, this will include your full peer review and any attached files.

.

Reviewer #1: No

Reviewer #2: **Yes:** Yuichi ObaYuichi Oba

---

## [Editor Report · Acceptance letter]

PONE-D-26-12952R1

PLOS One

Dear Dr. Ghedotti,

I'm pleased to inform you that your manuscript has been deemed suitable for publication in PLOS One. Congratulations! Your manuscript is now being handed over to our production team.

Kind regards,

on behalf of

Dr. Caio Santos Nogueira

Academic Editor

PLOS One